# Clinical Characteristics of Symptomatic Cholecystitis in Post-Gastrectomy Patients: 11 Years of Experience in a Single Center

**DOI:** 10.3390/medicina58101451

**Published:** 2022-10-14

**Authors:** Yun Suk Choi, Boram Cha, Sung Hoon Kim, Jin Wook Yi, Kyeong Deok Kim, Moon Suk Choi, Yoon Seok Heo

**Affiliations:** 1Department of Surgery, College of Medicine, Inha University Hospital, 27, Inhang-ro, Jung-gu, Inchon 22332, Korea; 2Department of Internal Medicine, College of Medicine, Inha University Hospital, 27, Inhang-ro, Jung-gu, Inchon 22332, Korea

**Keywords:** cholecystectomy, cholecystitis, gastrectomy

## Abstract

*Background and Objectives*: Gallbladder (GB) stones, a major cause of symptomatic cholecystitis, are more likely to develop in post gastrectomy people. Our purpose is to evaluate characteristics of symptomatic cholecystitis after gastrectomy. *Materials and Method*: In January 2011–December 2021, total 1587 patients underwent operations for symptomatic cholecystitis at our hospital. We reviewed the patients’ general characteristics, operation results, pathologic results, and postoperative complications. We classified the patients into non-gastrectomy and gastrectomy groups, further divided into subtotal gastrectomy and total gastrectomy groups. *Result*: The patients’ ages, male proportion, and the open surgery rate were significantly higher (127/1543 (8.2%) vs. 17/44 (38.6%); *p* < 0.001), and the operation time was longer (102.51 ± 52.43 vs. 167.39 ± 82.95; *p* < 0.001) in the gastrectomy group. Extended surgery rates were significantly higher in the gastrectomy group (56/1543 (3.6%) vs. 12/44 (27.3%); *p* < 0.001). The period from gastrectomy to symptomatic cholecystitis was significantly shorter in the total gastrectomy group (12.72 ± 10.50 vs. 7.25 ± 4.80; *p* = 0.040). *Conclusion*: GB stones were more likely to develop in post-gastrectomy patients and extended surgery rates were higher. The period to cholecystitis was shorter in total gastrectomy. Efforts to prevent GB stones are considered in post-gastrectomy patients.

## 1. Introduction

Gallbladder (GB) stones are the most common cause of symptomatic cholecystitis—one of the most common causes of abdominal emergency surgery—and of common bile duct stones that require endoscopic retrograde cholangiopancreatography (ERCP). The incidence of GB stones in the general population is known to be 2.2–5.0% [1,2], of which ~5% progress to symptomatic cholecystitis annually [3,4,5]. In Korea, >150,000 people are treated for cholelithiasis, and ~70,000 cholecystectomies are performed annually [6].

The reported incidence of GB stones after gastrectomy is 6.5–25%, which is higher than that in the general population [3,4,7,8]. The reasons for the high incidence of GB stones include hepatoduodenal lymph node dissection around the stomach, duodenum bypassing after gastrectomy, vagus nerve injury during gastrectomy, and rapid weight loss after surgery [4,9,10,11]. Due to these diverse causes, GB contraction decreases and bile salt concentration increases, leading to an increase in GB stone formation [12,13]. Several studies have recommended taking ursodeoxycholic acid (UDCA) to prevent post-gastrectomy GB stones and symptomatic cholecystitis [14,15,16].

Laparoscopic cholecystectomy was first performed in 1986 and is the gold standard for the treatment of symptomatic cholecystitis [17,18]. Postoperative adhesions can occur within the abdominal cavity in those who have undergone a gastrectomy, which can be particularly severe around the GB and in the Calot triangle due to hepatoduodenal ligament LN dissection. Therefore, cholecystectomy after gastrectomy is more difficult than cholecystectomy in the general population—the operation time is longer, the frequency of open cholecystectomy is higher, and the number of postoperative complications is higher [19,20]. The technique of laparoscopic surgery has been developed recently, and laparoscopic cholecystectomy is often performed even if there is a history of abdominal surgery [21,22].

Some investigators have attempted to perform prophylactic cholecystectomy together with gastrectomy, but this is not recommended [7,23,24,25]. The cholecystectomy is only performed at the same time as a gastrectomy if there is an abnormality within the GB before the preoperative evaluation. Although it is known that GB stones occur more frequently after gastrectomy, there are few reports on the association between GB stones and symptomatic cholecystitis. In addition, little is known about the differences in the clinical characteristics of symptomatic cholecystitis in patients with distal gastrectomy and total gastrectomy. In this study, we analyzed patients who visited the emergency room (ER) for symptomatic cholecystitis and underwent surgery at a single institution in South Korea over the past 11 years.

## 2. Materials and Methods

### 2.1. Data Collection and Patient Grouping

We retrospectively reviewed electric medical data for patients who visited the ER due to symptomatic cholecystitis at Inha University Hospital from January 2011 to December 2021. A total of 1587 patients were enrolled for the analysis. All patients were hospitalized for cholecystitis, and surgery was performed during hospitalization. We collected their clinical data, including patients’ general characteristics (age, sex, and body mass index [BMI, kg/m^2^]); personal history; operation-related variables such as the American Society of Anesthesiology [ASA] score; laboratory values—white blood cell (WBC), absolute neutrophil count (ANC), C-reactive protein (CRP), creatinine (Cr), hemoglobin (Hb), protein, albumin, bilirubin, aspartate aminotransferase (AST), alanine aminotransferase (ALT) and glucose—from the day of the operation to postoperative days 1 and 2; pathologic results, preoperative and postoperative clinical course, and medical and surgical complications related to cholecystectomy. Complications were evaluated as grades 0–V according to the Clavien–Dindo classification (CDC). We categorized the patients into two groups according to the CDC classification score: the mild complication group included CDC grades 0–III, and the severe complication group included CDC grades IV and V.

Patients who had undergone operations due to ulcer perforation, bariatric surgery, and other causes were excluded from the gastrectomy group. Only patients who underwent therapeutic gastrectomy for gastric cancer were included in the gastrectomy group. Since some patients underwent a gastrectomy at another hospital, all radiologic or endoscopic examinations before and after cholecystectomy were reviewed to evaluate the type of the previous gastrectomy.

### 2.2. Statistics and Ethics

Statistical analysis was performed using SPSS ver. 27.0 (SPSS Inc., Chicago, IL, USA). Either the chi-square test or Fisher’s exact test was used for cross-table analysis depending on the sample size. Unpaired *t*-tests were used to compare the means between two clinical groups. This study were approved by the Institutional Review Board of Inha University Hospital (IRB number: INH 2022-05-007).

## 3. Results

Among the 1587 patients enrolled in the study, 1543 had no history of gastrectomy and 44 had undergone gastrectomy (Table 1). In the gastrectomy group, the patients’ ages (57.58 ± 17.28 years vs. 66.98 ± 11.76 years; *p* < 0.001) and the proportion of men were significantly higher (802/1543 (52.0%) vs. 33/44 (75.0%); *p* = 0.003), while the patients’ BMIs were significantly lower than in the non-gastrectomy group (25.31 ± 3.99 vs. 21.66 ± 3.20; *p* < 0.001). Among 44 patients in the gastrectomy group, 35 had underwent subtotal gastrectomy and 9 underwent total gastrectomy. The average duration of symptomatic cholecystitis after gastrectomy was 11.54 years. The proportion of patients who had underwent percutaneous transhepatic gallbladder drainage (PTGBD) insertion was significantly higher in the gastrectomy group than in the non-gastrectomy group (147/1543 (9.5%) vs. 9/44 (20.5%); *p* = 0.016), but there was no significant difference in the PTGBD insertion period. There was no difference in the duration from cholecystitis diagnosis in the ER to operation between the two groups, but the postoperative hospital stay was significantly longer in the gastrectomy group (5.19 ± 6.89 days vs. 8.20 ± 4.87 days; *p* = 0.004).

Table 2 shows the pathologic and surgical results after surgery for the two groups. Among the patients diagnosed with cholecystitis on histological examination, the proportion of patients with severe cholecystitis showing gangrenous change, ulceration, and empyema as pathologic results was not different between the two groups (329/1519 (21.2%) vs. 11/43 (26.3%); *p* = 0.683). The rate of laparoscopic surgery was significantly lower in the gastrectomy group (1416/1538 (91.8%) vs. 27/44 (61.4%); *p* < 0.001). The proportion of patients who had undergone extended surgery for cholecystectomy combined with another surgery such as choledocolithotomy was significantly higher in the gastrectomy group (56/1543 (3.6%) vs. 12/44 (27.3%); *p* < 0.001), and the operation time was also significantly longer in the gastrectomy group (102.51 ± 52.43 min vs. 167.39 ± 82.95 min; *p* < 0.001). There was no difference in the postoperative surgical complication rates or the bile duct-related complications, which is one of the most significant complications, between the two groups. There was also no difference in the rates of postoperative medical complications, intensive care unit (ICU) hospitalizations, and mortality between the two groups. When the postoperative complications were classified by the Clavien–Dindo classification, there was no difference in the rates of postoperative severe complications between both groups.

A comparative analysis was performed between the two groups for laboratory tests performed on patients before and after surgery (Table 3 and Figure 1). Protein, albumin, and hemoglobin levels were significantly lower in the gastrectomy group before surgery. There was no difference in inflammatory markers such as WBC and CRP. However, bilirubin levels were higher in the gastrectomy group (1.67 ± 1.93 mg/dL vs. 2.44 ± 2.40 mg/dL; *p* = 0.055). On the postoperative day 1, WBC counts decreased in both groups, but the decrease was lower in the gastrectomy group, resulting in a significant difference between the two groups. Protein, albumin, and Hb levels were significantly lower in the gastrectomy group. Bilirubin was decreased in both groups, it was higher than normal in the gastrectomy group (1.09 ± 0.99 mg/dL vs. 1.51 ± 1.28 mg/dL; *p* = 0.052). On the postoperative day 2, WBC counts decreased in both groups. However, CRP levels continued to rise in the gastrectomy group, resulting in a significant difference between the two groups (11.80 ± 7.54 mg/L vs. 23.89 ± 1.35 mg/L; *p* = 0.026). Protein, albumin, and Hb levels were consistently and significantly lower in the gastrectomy group. Bilirubin levels were decreased in both groups, and there was no difference between the groups.

To avoid statistical error, we analyzed only male patients separately (Table 4). A total of 802 patients in the non-gastrectomy group and 33 patients in the gastrectomy group were included. The rate of laparoscopic surgery was significantly lower in the gastrectomy group (725/802 (90.4%) vs. 21/33 (63.6%); *p* < 0.001). The proportion of patients who had undergone extended surgery for cholecystectomy combined with another surgery such as choledocolithotomy was significantly higher in the gastrectomy group (35/802 (4.4%) vs. 7/33 (21.2%); *p* < 0.001), and the operation time was also significantly longer in the gastrectomy group (109.02 ± 54.54 min vs. 157.41 ± 75.21 min; *p* < 0.001). The results of analyzing only male patients were similar to those of analyzing all patients.

A total of 44 patients who underwent gastrectomy were divided into subtotal gastrectomy and total gastrectomy groups for comparative analysis (Table 5). There were no differences in age, sex, and BMI between the two groups. In addition, there were no differences in surgical approach, surgical extent, and the severity of cholecystitis between the groups. The period from gastrectomy to symptomatic cholecystitis was significantly shorter in the total gastrectomy group than in the subtotal gastrectomy group (12.72 ± 10.50 years vs. 7.25 ± 4.80 years; *p* = 0.040).

## 4. Discussion

If there is any abnormality in the GB during gastrectomy, cholecystectomy is usually performed alongside it. This is because the incidence rate of GB stones after gastrectomy is significantly higher than that in the general population, and cholecystectomy is more difficult after gastrectomy. However, it is a general treatment principle not to perform cholecystectomy in patients without any GB abnormality [7,23,24,25]. As the symptomatic cholecystitis patients in the gastrectomy group were older than the general population, the GB stones could have formed after the gastrectomy and led to the development of symptoms such as cholecystitis after a certain period. The relatively high proportion of male patients in the gastrectomy group seems to correlate with the relatively high incidence of gastric cancer in males. According to South Korea’s nationwide gastric cancer survey published in 2019, the prevalence of gastric cancer has approximately doubled in males over the past decade [26].

The proportion of patients who underwent gastrectomy among patients hospitalized in the ER for symptomatic cholecystitis was approximately 2.8%, which is similar to that reported in other studies [27,28]. It was determined that the gastrectomy did not affect the severity of inflammation or the pathology of the GB. Along to Tokyo guideline 2018, the severity of cholecystitis depends on the patient’s condition and laboratory data rather than the pathologic results, it was difficult to evaluate the relationship between the gastrectomy and the severity of cholecystitis in this study [29]. However, inflammatory markers such as WBC and CRP levels were higher, recovery in the laboratory findings was slower, and hospital stay was longer in the gastrectomy group. As previously reported, when cholecystectomy was performed in the gastrectomy group, the operation time was longer and the open surgery rate was higher [19,20]. This is attributed to the effect of severe adhesions forming after gastrectomy around the GB and the hepatoduodenal ligament. Cholecystectomy combined with choledocolithotomy was performed more frequently in the gastrectomy group (56/1543 (3.6%) vs. 12/44 (27.3%); *p* < 0.001). In this study, choledocolithotomy was also performed for CBD stone removal in 10 patients and for bile duct injury in 2 patients in the gastrectomy group. Roux-en-Y and Billroth II reconstructions in gastrectomy patients, excluding Billroth I, had much lower ERCP success rates than in the general population [30]. Because of the difficulty of ERCP, there is a possibility that the rate of choledocolithotomy was increased to complete the surgery in one stage.

In this study, it was found that the gastrectomy increased the difficulty of the cholecystitis surgery, as evidenced by the long operation times and high rates of open surgery, but did not affect the surgical complications. In addition, there was no difference in bile duct leak or stricture, one of the most important complications, between the two groups. In this study, eight patients in the non-gastrectomy group who had remnant CBD stones, bile duct leakage, and stenosis underwent ERCP after surgery. Five patients underwent endoscopic retrograde biliary drainage (ERBD), and three patients underwent endoscopic nasobiliary drainage (ENBD) to eliminate the bile duct-related complications. Considering that ERCP is difficult after gastrectomy, more attention should be paid to prevent biliary problems, particularly during surgery. There was no difference medical complications after surgery between the two groups, but the gastrectomy group had a longer hospital stay after surgery and delayed recovery of laboratory inflammation markers such as WBC and CRP levels. Thus, these findings suggest that more careful treatment is required for postoperative management in gastrectomy group.

In this study, the postoperative period for patients who underwent gastrectomy was approximately 11 years. The lower BMI and lower protein, albumin, and Hb levels of the patients appeared to be long-term nutritional effects after gastrectomy. Nutritional parameters such as protein, albumin, and Hb were lower than those of the general population, but they are all within the normal range. The gastrectomy group patients had an average BMI of 22 kg/m^2^, which was lower than that of the general cholecystitis patients. The BMI of gastrectomy group was within the normal range compared to general patients with grade I obesity. Although the average age of the patients in the gastrectomy group was higher, those patients had fewer severe complications, as evidenced by their CDC grades and ICU admission rates after surgery. There was no significant difference in the medical complication rates between general patients and the gastrectomy group. Thus, some nutritional deficits can occur after long-term gastrectomy, but they do not have much effect on the general condition of the patient.

In this study, it took an average of 11.13 years for patients to develop symptomatic cholecystitis after gastrectomy. In addition, the period from gastrectomy to symptomatic cholecystitis was approximately half in the total gastrectomy group (an average of 7.54 years) compared to the distal gastrectomy group (an average of 13.58 years). The posterior vagus nerve must be sacrificed after total gastrectomy. Due to the vagus nerve damage and food material bypassing the duodenum, GB contraction is reduced and bile concentration in the GB is increased. Therefore, GB stones are more likely to occur in total gastrectomy patients than in distal gastrectomy patients [4,11,13]. In addition, the proportion of patients who underwent total gastrectomy due to symptomatic cholecystitis was higher than the proportion of patients who underwent total gastrectomy for all types of gastric cancer [26]. Several studies have reported that GB stones are more likely to develop in patients who have undergone total gastrectomy than in those who have undergone distal gastrectomy [10,28,31]. Therefore, more attention should be paid to preventing GB stones after total gastrectomy.

## 5. Limitations

This study had several limitations. This was a retrospective review analysis, and the gastrectomy group contained a relatively small number of patients. Therefore, comparisons between the variables may also have limited statistical significance. There were inconsistencies because the surgeries were performed by various general surgeons who would also differ in their postoperative management practices; hence, these inconsistencies should be eliminated in future studies. Due to the small sample size, it was impossible to eliminate these differences in this study. A long-term follow-up prospective and large volume study on patients who underwent gastrectomy is needed to achieve better results.

## 6. Conclusions

The average duration from gastrectomy to surgery due to symptomatic cholecystitis was 11.13 years. In the case of total gastrectomy, this duration was shorter than that for distal gastrectomy. Endoscopic treatment such as ERCP is more difficult, and the rate of open surgery is high after gastrectomy. Therefore, efforts to prevent GB stones, such as UDCA, should be considered for post-gastrectomy patients, especially total gastrectomy patients.

## Figures and Tables

**Figure 1 medicina-58-01451-f001:**
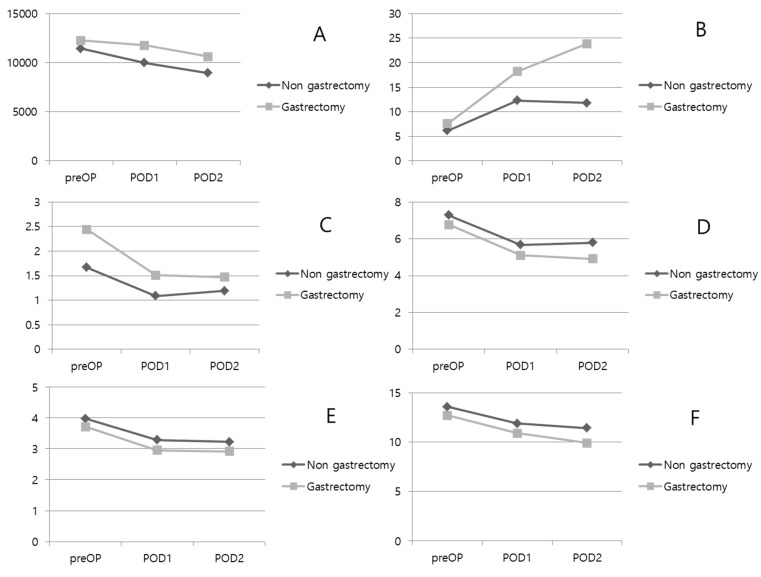
Perioperative laboratory results. (**A**): WBC. (**B**): CRP. (**C**): Total bilirubin. (**D**): Protein. (**E**): Albumin. (**F**): Hemoglobin. POD: postoperative day.

**Table 1 medicina-58-01451-t001:** Clinical characteristics of symptomatic cholecystitis patients.

Variable	Non-Gastrectomy (*n* = 1543)	Gastrectomy (*n* = 44)	*p*-Value
Age (years, mean ± sd)	57.58 ± 17.28	66.98 ± 11.78	<0.001
**Gender**			
Male	802 (52.0%)	33 (75.0%)	0.003
Female	741 (48.0%)	11 (25.0%)	
**ASA score**	2.54 ± 0.56	2.59 ± 0.58	0.650
I	18 (2.3%)	1 (3.4%)	
II	329 (42.2%)	10 (34.5%)	
III	428 (54.9%)	18 (62.1%)	
IV	5 (0.6%)	0 (0.0%)	
BMI (kg/m^2^)	25.31 ± 3.99	21.66 ± 3.20	<0.001
ICU admission days	0.17 ± 1.37	0.07 ± 0.26	0.638
Operation time (min, mean ± sd)	102.51 ± 52.43	167.39 ± 82.95	<0.001
**Stomach OP Hx**			
Subtotal gastrectomy		35 (79.6%)	
Total gastrectomy		9 (20.4%)	
**Stomach OP duration** (years, mean ± sd)	-	11.54 ± 9.40	
**PTGBD insertion**			
No	1396 (90.5%)	35 (79.5%)	0.016
Yes	147 (9.5%)	9 (20.5%)	
PTGBD insertion (days, mean ± sd)	8.17 ± 6.79	7.33 ± 2.60	0.715
**Post-op rescue (ERCP) procedure**			
No	1535 (99.5%)	44 (100.0%)	0.892
Yes	8 (0.5%)	0 (0.0%)	
Remnant CBD stone removal	3 (0.2%)		
Bile leak	4 (0.25%)		
Bile duct stenosis	1 (0.05%)		
From ER to operation (days, mean ± sd)	4.05 ± 4.55	4.61 ± 4.63	0.415
Postoperative hospital stay (days, mean ± sd)	5.19 ± 6.89	8.20 ± 4.87	0.004
Hospital stay (days, mean ± sd)	9.24 ± 8.92	12.82 ± 6.66	0.008

ASA: American Society of Anesthesiology, ASA I: A normal healthy patient, ASA II: A patient with mild systemic disease, ASA III: A patient with severe systemic disease, ASA IV: A patient with severe, life-threatening systemic disease, BMI: body mass index, kg/m^2^, ER: emergency room, NA: not applicable, PTGBD: percutaneous transhepatic gallbladder drainage, ERCP: endoscopic retrograde cholangiography, CBD: common bile duct.

**Table 2 medicina-58-01451-t002:** Pathologic and surgical characteristics of cholecystectomy patients.

Variable	Non-Gastrectomy (*n* = 1543)	Gastrectomy (*n* = 44)	*p*-Value
**Cholecystitis severity according to pathologic results**			
Gangrenous cholecystitis (gangrenous, ulceration, empyema)	329 (21.2%)	11 (26.3%)	0.683
Acute or chronic cholecystitis	1190 (78.4%)	32 (73.7%)	
**Pathologic details**			
Acute cholecystitis	462 (29.9%)	15 (34.1%)	NA
Chronic cholecystitis	728 (47.2%)	17 (38.6%)	
Gangrenous GB (gangrenous, ulceration, empyema)	329 (21.3%)	11 (25.0%)	
GB cancer	23 (1.5%)	1 (2.3%)	
Other	1 (0.1%)	0 (0.0%)	
**Operation time** (min, mean ± sd)	102.60 ± 52.82	164.87 ± 80.03	<0.001
**Surgical approach**			
Open surgery	127 (8.2%)	17 (38.6%)	<0.001
Laparoscopic surgery	1416 (91.8%)	27 (61.4%)	
**Surgical extent**			
Only cholecystectomy	1487 (96.4%)	32 (72.7%)	<0.001
Extended surgery	56 (3.6%)	12 (27.3%)	
**Surgical method details**			
Open cholecystectomy	93 (6.0%)	8 (18.2%)	NA
Laparoscopic cholecystectomy	1394 (90.3%)	24 (54.5%)	
OC + Choledocolithotomy	26 (1.7%)	9 (20.5%)	
LC + Choledocolithotomy	22 (1.4%)	3 (6.8%)	
Cholecystectomy with another operation (Small bowel resection, colon resection)	4 (0.3%)	0 (0.0%)	
Radical cholecystectomy	4 (0.3%)	0 (0.0%)	
**Post-op bile duct problem** **(leak, stricture)**			
No	1533 (99.4%)	44 (100.0%)	0.592
Yes	10 (0.6%)	0 (0.0%)	
**Surgical complication**			
No	1525 (98.8%)	44 (100.0%)	0.471
Yes	18 (1.2%)	0 (0.0%)	
**Type of surgical complication**			
Leak, fistula, perforation	12 (0.7%)	0 (0.0%)	N/A
Stricture, obstruction	1 (0.1%)	0 (0.0%)	
Ileus	1 (0.1%)	0 (0.0%)	
Bleeding	3 (0.2%)	0 (0.0%)	
Other	1 (0.1%)	0 (0.0%)	
**ICU admission due to a medical problem**			
No	1531 (99.2%)	44 (100.0%)	0.557
Yes	12 (0.8%)	0 (0.0%)	
**ICU admission days** (postoperative)	0.17 ± 1.37	0.07 ± 0.26	0.686
**Area of medical problem**			M/A
Lung	8 (0.5%)		
Renal	3 (0.2%)		
Infection	4 (0.3%)		
Cardio/Vascular	4 (0.3%)		
**Death**			
No	1531 (99.2%)	44 (100.0%)	0.557
Yes	12 (0.8%)	0 (0.0%)	
**Clavien–Dindo Classification**			
**Mild complication (I–III)**	1523 (98.6%)	44 (100.0%)	0.526
**Severe complication (IV–V)**	21 (1.4%)	0 (0.0%)	
IVa	7 (0.45%)	0 (0.0%)	
IVb	2 (0.12%)	0 (0.0%)	
V	12 (0.83%)	0 (0.0%)	

NA: not applicable, OC: open cholecystectomy, LC: laparoscopic cholecystectomy, ICU: intensive care unit.

**Table 3 medicina-58-01451-t003:** Perioperative laboratory results.

Variable	Non-Gastrectomy (*n* = 1543)	Gastrectomy (*n* = 44)	*p*-Value
**Pre-operative period**			
WBC	11.45 ± 5.22	12.27 ± 7.21	0.497
CRP	6.18 ± 8.80	7.57 ± 9.57	0.332
ANC	9216.85 ± 5194.38	10,541.47 ± 7030.879	0.262
Hb	13.58 ± 1.78	12.69 ± 1.84	0.003
Protein	7.28 ± 0.63	6.78 ± 0.76	<0.001
Albumin	3.98 ± 0.53	3.71 ± 0.44	0.003
Creatinine	1.02 ± 0.85	0.94 ± 0.39	0.566
Bilirubin	1.67 ± 1.93	2.44 ± 2.40	0.059
AST	120.20 ± 200.23	211.38 ± 278.12	0.055
ALT	116.84 ± 186.69	145.76 ± 158.89	0.283
**Post-operative day 1**			
WBC	9.98 ± 3.88	11.76 ± 5.17	0.043
CRP	12.32 ± 8.64	18.20 ± 9.63	0.248
ANC	8003.79 ± 3919.72	10,115.22 ± 5161.68	0.017
Hb	11.89 ± 1.72	10.88 ± 1.75	<0.001
Protein	5.67 ± 0.66	5.11 ± 0.74	<0.001
Albumin	3.28 ± 0.52	2.95 ± 0.41	<0.001
Creatinine	0.98 ± 1.00	0.78 ± 0.28	0.421
Bilirubin	1.09 ± 0.99	1.51 ± 1.28	0.052
AST	66.39 ± 78.61	91.00 ± 103.28	0.154
ALT	71.80 ± 86.91	88.84 ± 93.23	0.280
**Post-operative day 2**			
WBC	8.94 ± 3.99	10.62 ± 5.25	0.190
CRP	11.80 ± 7.54	23.89 ± 1.35	0.026
ANC	6726.37 ± 3922.23	8878.00 ± 5091.63	0.088
Hb	11.42 ± 1.95	9.91 ± 1.62	0.015
Protein	5.80 ± 0.76	4.91 ± 0.60	<0.001
Albumin	3.22 ± 0.51	2.92 ± 0.37	0.048
Creatinine	0.95 ± 0.84	0.67 ± 0.28	0.328
Bilirubin	1.19 ± 1.20	1.47 ± 1.07	0.428
AST	53.20 ± 79.83	63.00 ± 44.18	0.685
ALT	72.70 ± 83.51	77.73 ± 73.84	0.843

WBC: White Blood Cell count X 1000 (/µL),Hb: Hemoglobin (g/dL), ANC: Absolute Neutrophil Count (/µL), CRP: C-reactive protein (mg/L).

**Table 4 medicina-58-01451-t004:** Clinical and pathologic characteristics of male cholecystitis patients.

Variable	Non-Gastrectomy (*n* = 802)	Gastrectomy (*n* = 33)	*p*-Value
Age (years, mean ± sd)	58.66 ± 16.19	65.07 ± 13.11	0.042
ASA score	2.55 ± 0.55	2.47 ± 0.62	0.544
BMI (kg/m^2^)	25.50 ± 8.62	21.77 ± 2.57	0.025
**Stomach OP Hx**			
Subtotal gastrectomy		26 (78.8%)	
Total gastrectomy		7 (21.2%)	
**Stomach OP duration** (years, mean ± sd)	-	12.00 ± 10.48	
ICU admission days (days, mean ± sd)	0.25 ± 1.85	0.07 ± 0.27	0.612
Operation time (min, mean ± sd)	109.02 ± 54.54	157.41 ± 75.21	<0.001
Hospital stay (days, mean ± sd)	9.57 ± 9.15	11.22 ± 5.54	0.351
From ER to operation (days, mean ± sd)	4.18 ± 5.08	3.74 ± 3.21	0.653
Postoperative hospital stay (days, mean ± sd)	5.39 ± 6.31	7.48 ± 5.15	0.089
**Clavien–Dindo Classification**			
Mild complication (I–III)	789 (98.4%)	33 (100.0%)	
Severe complication (IV–V)	13 (1.6%)	0 (0.0%)	
**Cholecystitis severity** **according to pathologic results**			
Gangrenous cholecystitis (gangrenous, ulceration, empyema)	209 (26.1%)	8 (24.2%)	0.798
Acute or chronic cholecystitis	584 (72.8%)	25 (75.8%)	
**Pathologic details**			
Acute cholecystitis	242 (30.2%)	13 (39.4%)	NA
Chronic cholecystitis	342 (42.6%)	12 (36.4%)	
Gangrenous GB (gangrenous, ulceration, empyema)	209 (26.1%)	8 (24.2%)	
GB cancer	9 (1.1%)	0 (0.0%)	
**Surgical approach**			
Open surgery	77 (9.6%)	12 (36.4%)	<0.001
Laparoscopic surgery	725 (90.4%)	21 (63.6%)	
**Surgical extent**			
Only cholecystectomy	767 (95.6%)	26 (78.8%)	<0.001
Extended surgery	35 (4.4%)	7 (21.2%)	
**Surgical method details**			
Open cholecystectomy	54 (6.7%)	6 (18.2%)	NA
Laparoscopic cholecystectomy	713 (88.9%)	20 (60.6%)	
OC + Choledocolithotomy	20 (2.5%)	6 (18.2%)	
LC + Choledocolithotomy	12 (1.5%)	1 (3.0%)	
Cholecystectomy with another operation (Small bowel resection, colon resection)	1 (0.1%)	0 (0.0%)	
Radical cholecystectomy	2 (0.2%)	0 (0.0%)	

ASA: American Society of Anesthesiology, BMI: body mass index, kg/m^2^, ICU: intensive care unit, ER: emergency room, NA: not applicable, OC: open cholecystectomy, LC: laparoscopic cholecystectomy.

**Table 5 medicina-58-01451-t005:** Clinical characteristics according to subtotal gastrectomy and total gastrectomy.

Variable	Subtotal Gastrectomy (*n* = 35)	Total Gastrectomy (*n* = 9)	*p*-Value
**Age (years, mean ± sd)**	65.51 ± 12.15	64.89 ± 10.59	0.557
**Gender**			
Male	25 (71.4%)	8 (88.9%)	0.281
Female	10 (28.6%)	1 (11.1%)	
**BMI** (kg/m^2^)	21.76 ± 3.48	21.27 ± 1.81	0.572
**Previous gastrectomy type**			
Subtotal gastrectomy, Billroth I	11 (31.4%)		
Subtotal gastrectomy, Billroth II	19 (54.3%)		
Subtotal gastrectomy, Roux-en-Y	4 (11.4%)		
Proximal gastrectomy	1 (2.9%)		
Total gastrectomy, Roux-en-Y		9 (100%)	
**Surgical approach**			
Open	13 (37.1%)	4 (44.4%)	0.688
Laparoscopic	22 (62.9%)	5 (55.6%)	
**Surgical extent**			
Only cholecystectomy	26 (74.3%)	6 (66.7%)	0.647
Extended surgery	9 (25.7%)	3 (33.3%)	
**ASA score**	2.67 ± 0.48	2.20 ± 0.83	0.095
**Pathologic diagnosis**			
Gangrenous cholecystitis (gangrenous, ulceration, empyema)	9 (25.7%)	3 (33.3%)	0.647
Acute or chronic cholecystitis	26 (74.3%)	6 (66.7%)	
Operation time (min, mean ± sd)	169.57 ± 80.03	158.89 ± 98.29	0.735
**Gastrectomy duration** (years, mean ± sd)	12.72 ± 10.5	7.25 ± 4.80	0.040
From ER to operation (days, mean ± sd)	4.97 ± 4.99	3.22 ± 2.63	0.159
Postoperative hospital stay (days, mean ± sd)	8.43 ± 5.26	7.33 ± 3.00	0.554
Hospital stay (days, mean ± sd)	13.40 ± 7.34	10.56 ± 1.59	0.258

ASA: American Society of Anesthesiology, BMI: body mass index, kg/m^2^, ER: emergency room, NA: not applicable.

## Data Availability

Data sharing not applicable.

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
