# Peer review of "Clinical Characteristics of Symptomatic Cholecystitis in Post-Gastrectomy Patients: 11 Years of Experience in a Single Center"

_medicina, 2022, doi:10.3390/medicina58101451_

Round 1

Reviewer 1 Report

The manuscript is well written. The project has clinical significance. A major concern is that the sample size in the gastrectology group (n=44) is too small to draw a solid conclusion. In addition, sex is a significant contributing factor to the disease, and there is a sex disparity in the small group (33 males and 11 females). I suggest that authors may  compare the males in the non-gastrectomy group (n=802) with the males in the gastrectomy group (n=33). Although in this case the sample size is still small with n=33, the comparison would eliminate the sex factor. 

Author Response

Reviwer 1

The manuscript is well written. The project has clinical significance. A major concern is that the sample size in the gastrectology group (n=44) is too small to draw a solid conclusion. In addition, sex is a significant contributing factor to the disease, and there is a sex disparity in the small group (33 males and 11 females). I suggest that authors may compare the males in the non-gastrectomy group (n=802) with the males in the gastrectomy group (n=33). Although in this case the sample size is still small with n=33, the comparison would eliminate the sex factor. 

We sincerely appreciate your valuable comments. As your opinion, analyzing only male patients can rule out errors caused by sex factors, and I think it is good advice if you think that stomach cancer occurs more in male.

We analyzed 802 non-gastrectomy groups and 33 gastrectomy groups and we made Table 4. And the result was added to result. The results did not differ from the analysis of the entire patient.

Added content is indicated in red text.

Sincerely for your review.

Reviewer 2 Report

Congratulation for the work.

The abstract is not appealing for the reader and the conclusion has no new information.

Kindly ask you to clarify the gap that this study fills.

Author Response

We really appreciate your comments.

We will work harder in the future to write better research and paper.

Once again, thank you very much.

Round 2

Reviewer 1 Report

My concern has been addressed. 

Author Response

Sincere thanks for your review.

Thanks to you, this paper can become a better paper.

We will work harder to write better research and papers.

Thanks again for your hard work.

Reviewer 2 Report

At first round i just asked you to upgrade the abstract. I am really sorry that the way i wrote it, created a comfusion. 

I do believe, that minor changes (about the importance of the study) in introduction and conclusion of the abstract will upgrade it.

Author Response

Sincere thanks for your review.

Thanks to you, this paper can become a better paper.

Based on your review, we have revised the abstract as follows. The revised content is indicated in red text. Please understand that the abstract word limit is limited to 200.

We will work harder to write better research and papers.

Thanks again for your hard work.
